# Vitamin D, Cholesterol, and DXA Value Relationship with Bimaxillary Cone Beam CT Values

**DOI:** 10.3390/jcm12072678

**Published:** 2023-04-03

**Authors:** Mohammed G. Sghaireen, Kiran Kumar Ganji, Kumar Chandan Srivastava, Mohammad Khursheed Alam, Shadi Nashwan, Fayeq Hasan Migdadi, Ahmad Al-Qerem, Yousef Khader

**Affiliations:** 1Department of Prosthetic Dentistry, Jouf University, Sakaka 72388, Saudi Arabia; 2Department of Preventive Dentistry, College of Dentistry, Jouf University, Sakaka 72345, Saudi Arabia; mkalam@ju.edu.sa; 3Department of Oral & Maxillofacial Surgery & Diagnostic Sciences, College of Dentistry, Jouf University, Sakaka 72345, Saudi Arabia; kchandan@ju.edu.sa; 4Department of Computer Science, College of Computer and Information Sciences, Jouf University, Sakaka 72341, Saudi Arabia; 5Al Dhafra Hospitals, Abu Dhabi P.O. Box 50018, United Arab Emirates; migdadifayeq@gmail.com; 6Department of Computer Science, Faculty of Information Technology, Zarqa University, Zarqa 13110, Jordan; ahmad_qerm@zu.edu.jo; 7Department of Public Health, Jordan University of Science & Technology, Ar-Ramtha 3030, Jordan; yskhader@just.edo.jo

**Keywords:** computed tomography, cholesterol, cone beam computed tomography, osteoporosis, vitamin D

## Abstract

We evaluated the correlation that Vitamin D (Vit D), cholesterol levels, and T- and Z-scores of dual-energy X-ray absorptiometry (DXA) scans have with cone beam computed tomography values assessed in the anterior and posterior regions of maxillary and mandibular jaws. In total, 187 patients were recruited for this clinical study. Patients’ ages ranged between 45 and 65 years. Patients with valid DXA results, serum Vit D and cholesterol levels, and no evidence of bone disorders in the maxilla or mandibular region were included in the study and grouped in the control (non-osteoporosis) and case (osteoporosis) groups. Patients with a history of medical or dental disease that might complicate the dental implant therapy, chronic alcohol users, and patients who took calcium or Vit D supplements were excluded. The outcome variables assessed in the investigation were Vit D, cholesterol, Z-values, and cone beam computed tomography values. Regarding the case group, a significant (*p* < 0.05) inverse relationship was observed between Vit D and cholesterol. Although insignificant (*p* > 0.05), a positive relationship was found between Vit D and the cone beam computed tomography values in all regions of the jaws, except the mandibular posterior region (*p* < 0.05). Pearson correlation analysis was carried out. Vit D and cholesterol showed a statistically insignificant (*p* > 0.05) negative association with the cone beam computed tomography values in all regions of the jaws. However, the Z-values were highly correlated with the cone beam computed tomography values in all regions of the jaws (r > 7, *p* < 0.05). Vit D, cholesterol levels, and Z-values in women and men from young adulthood to middle age (45–65) were related with the cone beam computed tomography values of the jaws.

## 1. Introduction

Vitamin D (Vit D) is a hormone that regulates calcium phosphate homeostasis and mineral bone metabolism. Our body mainly obtains Vit D from three sources: skin, liver, and kidney, where 80–90% of the sources is sunlight through the skin, and the remaining 10–20% are dietary sources [1]. Vit D is a fat-soluble biomolecule originating from a Vit D-rich diet and sunlight exposure. Vit D3 (1, 25-Dihydroxy VD3) in its active form stimulates the intestinal absorption of calcium and phosphate; thereby, bone mineral homeostasis occurs. The activation of bone-forming osteoblasts and bone-resorbing osteoclast cells thus regulates bone mineralization. Thus, Vit D supplementation is the treatment of choice in preventing and treating osteoporosis [2,3].

Vit D is an important factor during osseointegration and implant stability. It establishes several bone metabolisms by regulating the gene expression of osteocalcin, osteopontin, albindin, and 24-hydroxylase. It increases extracellular matrix protein formation by osteoblasts. It stimulates osteoclast activity, thus inducing bone formation and improving the immune response, since all immune system cells contain Vit D receptors. This vitamin is essential for an antibacterial response, as it influences monocyte–macrophage reaction and inhibits the expression of certain pro-inflammatory cytokines. Vit D deficiency corresponds to a serum level below 20 µg/L; extreme deficiency is considered below 10 µg/L. Deficient Vit D diminishes early implant stability and increases the chance of dental implant failure [3,4]. Tania, W., et al., in 2011, concluded that supplementing Vit D to children and adolescents with Vit D deficiency could improve bone mineral density [5]. However, in recent meta-analyses, it was found that treatment with Vit D has no significant value in treating osteoporosis or fractures [6,7]

Cholesterol and Vit D levels have the same precursor, called dehydrocholestrol. Elevated low-density lipoprotein causes an imbalance in bone remodeling and a reduction in bone mass, and it is associated with an inflammatory response to bacteria. Certain enzymes released from bone are involved in the oxidation of low-density lipoprotein. Accumulated oxidized low-density lipoprotein exerts a deleterious effect on bone density and may cause implant failure, and it causes cell death by inhibiting bone-forming osteoblast cells. Several antioxidants in high-density lipoprotein interrupt the oxidation of low-density lipoprotein and are thus considered bone cell protectors and good cholesterol [8].

Various techniques are available for measuring bone mineral density in different points along the skeleton. Bone density is measured using T-scores compared with those of a healthy individual, whereas Z-scores are compared among individuals of the same age, gender, and size. Experts prefer Z-scores in children, teenagers, premenopausal females, and younger males. They help diagnose secondary osteoporosis, which results from underlying medical conditions, rather than primary osteoporosis. A T-score or Z-score is calculated once the cone beam computed tomography value has been measured by the manufacturer’s software. T-scores and Z-scores are calculated on a standard deviation scale by comparing the results to those of a reference population. Reference groups for T-scores should be young, gender-matched populations at peak bone mass, while for Z-scores, aged groups should be used. Several scientific publications refer to T-scores and Z-scores to determine osteoporosis and bone mineral density. These values are used for dual-energy X-ray absorptiometry (DXA) diagnostic criteria and current osteoporosis management guidelines [9].

It is believed that serum cholesterol levels and cholesterol-lowering drugs influence the bone mineral density of an individual. Cholesterol levels and bone mineral density are inversely correlated, according to Yuchen Chang [10]. Clinicians should pay close attention to female patients with high HDL-C levels, which may indicate osteoporosis or osteopenia risk. Early intervention is recommended in these patients, and so is closely monitoring their cone beam computed tomography values. High cholesterol increases osteoclast activity and osteoblast function [11]. This study evaluated the correlation that Vit D, cholesterol levels, and T- and Z-scores of DXA scans have with cone beam computed tomography values assessed in the anterior and posterior regions of the maxillary and mandibular jaws.

## 2. Materials and Methods

### 2.1. Study Design and Characteristics

Patients’ age ranged between 45 and 65 years. The patients attended the prosthodontics clinic for dental implant treatment. Data were collected between March 2019 and March 2021. The Local Committee for bioethics, Jouf University approved this cross-sectional study wide reference number 381-41. All patients were asked to sign their informed consent before participating in this study. The principles of the Helsinki Declaration (9th version, 2013) were followed throughout this clinical inquiry. In the current study, we included patients in the age group of 45–65 years with valid DXA results, serum Vit D, and cholesterol levels. Specifically, patients had no evidence of bone disorders in the maxilla or mandibular region; they had no history of chronic thyroid disease nor diabetes, acute nor chronic inflammatory disorders, heart disease, liver disease, nor kidney disease; they had not used steroidal medications such as denosumab, prednisolone, dexamethasone, or betamethasone within the previous six months; they had not used hormone therapy within the previous two years and had not used anti-osteoporotic drugs (raloxifene or bisphosphonates). Before enrollment, informed consent was obtained for willingness to participate in this study. Patients with a medical or dental disease history that might complicate the dental implant therapy, chronic alcohol users, and patients taking calcium or Vit D supplements were excluded from this study. The diagnosis of osteoporosis was made based on reference databases used by DXA device manufacturers for T-score reference recommended by National Health and Nutrition Examination Survey III databases for the hip and lumbar spine [12]. Based on inclusion and exclusion criteria, the selected participants were recruited into the study using a convenient sampling technique. Then, they were allocated to a control group (non-osteoporosis) and a case group (osteoporosis) using a simple random probability technique. T- and Z-values, and cholesterol levels were recorded from patients’ DXA charts, and Vit D was recorded from patients’ medical records. The normal cholesterol range in males and females was 125 to 200 mg/dL, and the normal range of Vit D in males and females was 30–70 ng/mL. Cone beam computed tomography value data were collected from the patient’s cone bean computed tomography records (SORDEX; Nahkelantie 160 Tuusula, Finland), which were recorded as a part of the diagnostic protocol before implant therapy. According to a previous study, all cone beams computed tomography scans were taken by the same operator, and all the bone density measurements were collected by the same investigator (M.S.) [13]. The recruited patients’ cone beam computed tomography values were recorded in four regions, i.e., maxillary anterior, mandibular anterior, mandibular posterior, and maxillary posterior, in the case and control groups. Cone beam computed tomography values were recorded 3 mm apical to the alveolar crest in the central incisor and at the first molar of the maxilla and mandible jaws. The researchers involved in the data collection from these patients were blinded and coded by the principal investigator and were unaware of group allocation. Later, after result analysis, the data were decoded for inferential statistics.

### 2.2. Sample Size Estimation and Outcome Variables

Sample size was estimated using the University of California San Francisco online sample size calculator tool. The total sample size required to determine whether a correlation coefficient differed from zero was estimated using type I rate error (α = 0.05; two-tailed), type II error (β = 0.20; considering the probability of failing to reject the null hypothesis under the alternative hypothesis), and expected correlation coefficient (*r* = 0.20). Hence, in total, 187 patients were recruited for this clinical study. Intra-examiner reliability was confirmed, and the kappa value was found satisfactory. The kappa value was calculated by recalculating 15 cases with one-week intervals.

### 2.3. Statistical Analysis

The gathered data were entered into a Microsoft Excel sheet. The Kolmogorov–Smirnov normality test was used to assess the normality of the data. Considering the data’s normal distribution and quantitative nature, the parametric test was used for inferential analysis. An unpaired *t*-test was employed to compare parameters, including baseline variables, serum Vit D, cholesterol, cone beam computed tomography values, and Z-values, between the study groups. Pearson correlation analysis was conducted on variables showing significance in inferential analysis. All statistical data analyses were performed using version 21 of Statistical Package for the Social Sciences (SPSS; IBM, and Chicago, IL, USA).

## 3. Results

Table 1 compares baseline variables between the study groups using an independent t-test and a chi-squared test at a 95% confidence interval. None of the variables showed any statistically significant (*p* > 0.05) difference between the study groups, suggesting that no confounding factors influenced the study results.

Table 2 compares the biochemical variables between the study groups; an independent t-test was applied at a 95% confidence interval. The Vit D levels were significantly (*p* < 0.01) higher in the control group than in the case group. On the contrary, the cholesterol level was significantly (*p* < 0.05) elevated in the case group compared with the subjects in the control group.

Table 3 compares cone beam computed tomography values in different jaw regions between the study groups using an independent *t*-test at a 95% confidence interval. All regions of the jaw had significantly (*p* = 0.000) higher cone beam computed tomography values in the control group than in the case group. The present results reconfirm the DXA results, considered the basis for patient selection as osteoporosis and healthy subjects.

Table 4 shows the comparison of Z-values between the study groups. An independent t-test was applied at a 95% confidence interval. The Z-values were significantly (*p* = 0.000) higher in the control group than in the case group.

Table 5 shows the correlation analysis of the parameters of the control group; Pearson correlation was applied at a 95% confidence interval. Vit D showed a statistically significant (*p* < 0.05) positive correlation with the cone beam computed tomography values in all the regions of the jaws. Conversely, cholesterol demonstrated a significant negative correlation with the cone beam computed tomography values in all regions of the jaws (*p* < 0.05), except for the maxillary anterior region (*p* > 0.05). The Z-values also had a significant (*p* < 0.001) positive correlation with all regions’ cone beam computed tomography values.

Table 6 shows the correlation analysis of the parameters of the case group. Pearson correlation was applied at a 95% confidence interval. A significant (*p* < 0.05) inverse relationship was observed between Vit D and cholesterol. Although insignificant (*p* > 0.05), a positive relationship was found between Vit D and the cone beam computed tomography values in all regions of the jaws, except the mandibular posterior region (*p* < 0.05). Unlike Vit D, cholesterol showed a statistically insignificant (*p* > 0.05) negative association with the cone beam computed tomography values in all regions of the jaws. However, the Z-values were highly correlated with the cone beam computed tomography values in all regions of the jaws (*p* < 0.05).

## 4. Discussion

Among the essential and critical vitamins linked to bone growth hormones is Vit D. It is also important for reducing inflammation and improving the body’s natural immune responses [14]. Additionally, cholesterol and Vit D have a close relationship. 7-Dehydrocholesterol, cholesterol’s precursor, is similar to Vit D [15]. Normally, there is good cholesterol (high-density lipoproteins) and bad cholesterol (low-density lipoproteins) [16].

A steroid hormone, Vit D, can be acquired from food or synthesized by the skin through sunlight exposure (ultraviolet light) [17]. Previtamin D3 is synthesized from cholesterol and isomerized to Vit D. In the liver, CYP27A1 enzymatically hydroxylates vitamin D3 to produce 25-hydroxyvitamin D3 (calcidiol or 25OHD3) after binding to Vit D-binding carrier protein [18]. It is known that Vit D stimulates osteoclast activity and the production of extracellular matrix proteins by osteoblasts, which are among the effects of Vit D on bone. Vit D deficiency in these patients is primarily responsible for catabolized bone turnover, resulting in osteoporotic fractures due to bone catabolism [18]. Clinical trials and patients suffering from fractures have also linked Vit D deficiency to impaired fracture healing [19]. The steroid hormone plays a key role in the regeneration of bone in this study.

The current investigation is the first one to correlate Vit D, cholesterol, Z-values, and the cone beam computed tomography values of the jaws. Our model of the relationship presented that Vit D had a positive correlation with the cone beam computed tomography values. Conversely, cholesterol demonstrated an inverse correlation with the cone beam computed tomography values, and the Z-values positively correlated with the cone beam computed tomography values in all regions of the jaws. One possible explanation for this challenging correlation is epigenetic modification [20]. Epigenetic modifications could explain the interaction between cone beam computed tomography values and cholesterol levels [21,22], as evidence suggests that both factors are affected by genetic and environmental factors. Vit D may mediate the relationship between the lipid profile and the bone status based on a review of biochemical mechanisms. More than 0.5% of the human genome has Vit D response elements, according to genetic studies [23]. Vit D, directly or due to epigenetic contributions, controls target genes’ gene expression and function [23]. In Vit D-deficient rats, peri-implant bone formation was decreased, as the osseointegration of dental implants is also dependent on bone regeneration [24]. According to Dvorak et al., a Vit D-rich diet can compensate for Vit D deficiency’s negative impact on cortical peri-implant bone formation in ovariectomized rats. Several allergic disorders and immune system dysfunction have also been linked to Vit D [25]. Immunomodulation is one of its major effects. Researchers now know that the immune system contains all the characteristics required to convert 25-hydroxy Vit D into active 1,25-dihydroxy Vit D during bacterial infection [14]. Bone mineral density is influenced by Vit D in a significant way and is a promising candidate for preventive interventions, such as the systemic supplementation of dairy products. Low bone mass and increased bone remodeling have been associated with Vit D deficiency in a postmenopausal population otherwise healthy [26]. It is necessary and recommended for patients undergoing dental implants or bone grafts to assess their total serum cholesterol levels and Vit D status. This study found that Vit D, cholesterol, and *Z*-values had a positive correlation with the cone beam computed tomography values in the anterior and posterior regions of maxillary and mandibular jaws. The present results reconfirm the DXA results, which were considered the basis for patient selection as osteoporosis and healthy subjects. These findings agree with recent investigations [27,28,29]. The associations between cone beam computed tomography values at four different sites, and Vit D, cholesterol, and Z-values were significant, though not large. That such associations were apparent was surprising, given the large measurement errors inherent in this study. Even in this heterogeneous group and despite the likely measurement errors, the consistency of the findings with different measures of bone density and stepwise relationships with Vit D, cholesterol, and *Z*-values as associated biochemical variables suggests a true association. The concomitant biochemical relations indicate that this association is real. With the present result, in terms of the direction of the association, it can be understood that with the rise in available Vit D, there is an increase in cone beam CT values. Cholesterol being a precursor of Vit D shows an inverse relation.

Another study found that Vit D levels below 30 ng/mL (75 nmol/L) were strongly related with inadequate calcium absorption. In studies on fracture incidence, Vit D levels above 75 nmol/L were found to reduce fracture incidence because of the beneficial effects on bone metabolism and muscle strength [30]. The current study’s findings agree with Maghbooli, Z., et al.’s [31] investigation, which proposed that Vit D deficiency and serum cholesterol levels have a negative correlation with bone status in the case of postmenopausal women. Findings from the current investigation indicate that the Z-values significantly impacted the cone beam CT values of the jaws irrespective of the region. The correlation of Z-values and cone beam CT values was independent of the participants’ systemic status, as their relationship remained the same in both groups. The findings of the current investigation also support the hypothesis that Vit D, cholesterol, and Z-values influence the bone density of the jaws in middle-aged individuals (45–65 years). The results re-emphasize the positive influence of Vit D on bone metabolism. Thus, osteoporosis patients showed marked reduced Vit D compared with normal subjects. Likewise, the role of cholesterol, the precursor of Vit D, was elevated in the case group, as it remained unutilized. Osteoporosis plays a significant role in this study, and we should not ignore its significance. Menopausal status, diuretics, and postmenopausal hormone replacement therapy were not associated with these associations, despite possible confounding factors, including age, body mass index, and smoking habit.

Since this was a cross-sectional study, there were some limitations: Firstly, there was no significant correlation among cholesterol, Vit D, and the cone beam computed tomography values of the jaws. The present investigation, however, is the first one to explain conflicting Vit D and cholesterol levels when *Z*-values are considered. In addition, not all age groups were included in the distinction. To reduce the effects of confounding factors, this relationship was only studied in the middle-age group. A single sample was used to measure the Vit D serum levels. An individual’s true Vit D level may not be reflected in the blood after a single measurement. Lastly, the design of our study was cross-sectional, so we could not explain the cause-and-effect relationship. More large-scale studies and clinical trials investigating the interplay among serum Vit D, lipids, and cone beam computed tomography values could help unravel its complex mechanism. Several strengths were identified in this study. Several repeated measurements of cone beam computed tomography values and potential confounders were taken over a long period. Furthermore, a population-based sample of women and men was used to measure cone beam computed tomography values at several bone sites. Women and men consistently found null results across all bone sites. Vit D, cholesterol, and *Z*-values lacking an association with cone beam computed tomography values is unlikely to be explained by insufficient study power or measurement error if the sample size is large and the number of repeated measures is high.

### Limitations

Statistical significance and clinical significance are important notations, as demonstrated in any scientific research approach. It is possible for statistically significant results not to have clinical significance, and vice versa [32]. Researchers should look beyond the threshold “*p*”-value in evaluating research results from a clinical standpoint instead of just considering “*p*”-values to determine worth. Hence, the findings of the current study limit application to the level of statistical significance only, and more randomized clinical trials are needed to confirm its clinical relevance. During the process of analyzing the results of the study, researchers should also take into account the study design, sample size, effect size, incorporated bias, and reproducibility. Hence, researchers with logical and critical thinking minds are best positioned to evaluate the research results and apply them into practice. Overall, the relationship model employed in the current study provides relevant information that invites to be cautious about the level of osseointegration before implant placement in subjects compromised with osteoporotic conditions.

## 5. Conclusions

Vit D, cholesterol levels, and *Z*-values in women and men from young adulthood to middle age (45–65 years) were related with the cone beam computed tomography values of the jaws. The established relationship provides important information to the implantologist as an alarming message to avoid complications following implant therapy. However, more randomized clinical trials are needed to confirm whether this relationship is clinically significant.

## Figures and Tables

**Table 1 jcm-12-02678-t001:** Comparative analysis of baseline characteristics between the study groups.

Variable		Study Group		*p*-Value
		Control Group (*n* = 90)	Case Group (*n* = 97)	
Age		55.5 ± 9.7	59.6 ± 8.2	0.37
Gender	Male	40 (45%)	42 (44%)	0.22
	Female	50 (55%)	55 (56%)	

**Table 2 jcm-12-02678-t002:** Comparative analysis of biochemical variables between the study groups.

Variable	Study Group		*p*-Value
	Control Group (*n* = 90)	Case Group (*n* = 97)	
Vit D	17.4 ± 5.2	13.6 ± 5.0	*p* < 0.01
Cholesterol	233.69 ± 24.950	247.73 ± 29.962	*p* < 0.05

**Table 3 jcm-12-02678-t003:** Comparative analysis of cone beam computed tomography value variables between the study groups.

Variable	Study Group		*p*-Value
	Control Group (*n* = 90)	Case Group (*n* = 97)	
Anterior mandible	897.67 ± 143.161	533.40 ± 83.633	*p* < 0.001
Posterior mandible	739.87 ± 178.623	351.77 ± 52.170	*p* < 0.001
Anterior maxilla	753.62 ± 186.895	381.58 ± 48.578	*p* < 0.001
Posterior maxilla	548.03 ± 110.584	149.06 ± 73.257	*p* < 0.001

**Table 4 jcm-12-02678-t004:** Comparative analysis of Z-values between the study groups.

Variable	Study Group		*p*-Value
	Control Group (*n* = 90)	Case Group (*n* = 97)	
Z-Value	0.241 ± 0.8565	−2.996 ± 0.4510	*p* < 0.001

**Table 5 jcm-12-02678-t005:** Correlation analysis of parameters of the control group.

	Vit D	Cholesterol	Z-Value	Cone Beam Computed Tomography Value (Anterior Mandibular)	Cone Beam Computed Tomography Value (Posterior Mandibular)	Cone Beam Computed Tomography Value (Anterior Maxillary)	Cone Beam Computed Tomography Value (Posterior Maxillary)
Vit D	-	0.112 (−0.269)	0.153 (0.243)	0.003 ** (0.428)	0.001 ** (0.526)	0.001 ** (0.530)	0.037 * (0.348)
Cholesterol	0.112 (−0.269)	-	0.086 (−0.279)	0.007 ** (−0.425)	0.018 * (−0.378)	0.086 (−0.279)	0.021 * (−0.369)
Z-value	0.086 (−0.279)	0.153 (0.243)	-	0.000 *** (0.751)	0.000 *** (0.774)	0.000 *** (0.715)	0.000 *** (0.787)

* *p* < 0.05, ** *p* < 0.01, *** *p* < 0.001.

**Table 6 jcm-12-02678-t006:** Correlation analysis of parameters of the case group.

	Vit D	Cholesterol	Z-Value	Cone Beam Computed Tomography Value (Anterior Mandibular)	Cone Beam Computed Tomography Value (Posterior Mandibular)	Cone Beam Computed Tomography Value (Anterior Maxillary)	Cone Beam Computed Tomography Value (Posterior Maxillary)
Vit D	-	0.042 (−0.294)	0.315 (0.148)	0.194 (0.191)	0.044 (0.292)	0.066 (0.268)	0.743 (0.049)
Cholesterol	0.042 (−0.294)	-	0.272 (−0.162)	0.782 (−0.041)	0.751 (−0.047)	0.309 (−0.150)	0.177 (−0.198)
Z-value	0.315 (0.148)	0.272 (−0.162)	-	0.000 (0.652)	0.000 (0.587)	0.000 (0.621)	0.000 (0.449)

Results are expressed as *p*-values (Pearson correlation coefficient).

## Data Availability

Data will be made available upon request to the corresponding author.

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
