# Peer review of "Vitamin D, Cholesterol, and DXA Value Relationship with Bimaxillary Cone Beam CT Values"

_jcm, 2023, doi:10.3390/jcm12072678_

Round 1

Reviewer 1 Report (New Reviewer)

-In the abstract, the p-value should be accompanied by the identification of the statistical test;

- Statistical p must be written in lower case;

- Line 56: “et al”. and line 145: “between the”;

- In Table 1, the distribution of percentages by gender in the control group needs to be

rectified;

- The discussion of the results should only be carried out in the discussion section (for

example the sentences of the lines 155-158, 164-166, 178-181, 190-193);

- It should be written CBCT or spelled out (Cone Beam Computed Tomography);

- The type of study should be mentioned in the materials and methods (it is only

mentioned at the end of the discussion);

- What was the result of calculating the sample size? Was it 187?;

- The division between control group and test group was random, but what was the

method of initial selection of subjects? Was it random?;

- What were they based on to define the age criterion?;

- What were they based on to define normal values for cholesterol and vitamin D?;

- Applied normality test should be mentioned;

- The conclusion needs to be more detailed.

Author Response

Dear Respected Reviewer,

Thanks for providing us with your valuable comments. All the comments were addressed using point to point clarification in the attached file.

Thanking you

Regards

Reviewer 2 Report (New Reviewer)

The paper has well-designed research methods, appropriate statistical analysis and a relatively good interpretation of the results. The results of the research are not generalizable, the conclusion is in accordance with the objectives of the research, its results and their interpretation, as well as the relevant literature.

The quality of the language in the context of grammar, syntax and spelling is solid, however, when reading the paper I occasionally get a certain impression of confusion. Certainly, the manuscript is written in a way that shows that the author has considered different perspectives and thoughtfully evaluated the evidence and arguments he presented.

The idea is original and differs from previous papers on a similar topic, i.e. the paper brings new knowledge and contributes to the development of the research area.

The manuscript needs some minor improvements; there are a few suggestions that authors may consider to improve it further.

Please accord keywords to Mesh word

You need to review the grammar of your paper, I suggest to use Grammarly.

Please add a table with the list of abbreviations used in the text.

I suggest to add a Section to add all the limitation of the study

Conclusions must be improved.

Author Response

Dear Respected Reviewer,

Thanks for providing us with your valuable comments. All the comments were addressed using point to point clarification in the attached file.

Thanking you

Regards

Reviewer 3 Report (New Reviewer)

Dear Editorial Team, Thank you for the honor of reviewing this paper. After a thorough analysis of the paper, I conclude that it might be of high interest to the readers of your journal. I see the most urgent need for improvement in terms of language. I ask the authors to seek advice from a native speaker. After establishing adequate readability, the content of the text can be discussed. Enclosed are some corrections for the first two chapters.

Line 25: Change „patients who“ to “patients which”.

Line 32-34: please remove the last scentence of the abstract as it can’t be concluded from the data.

Line 42: Please remove the comma.

Line 48: Please change “osteoclacin” to “osteocalcin”.

Line 61: Please change “level has same precursor” to “have the same precursor”.

Line 62: Cause(s).

Line 64: Change “involves in” to “are involved in”.

Line 66-68: Please check gramma.

Line 69: Change “a varity of techniques are available” to “a varity of techniques is available”.

Line 74: Change “are” to “is”.

Introduction: Please explain what T- and Z-Scores exactly are.

Line 101: Please add “denosumab” to your list.

Line 135: Change “2.2. Statistical analysis” to “2.3. Statistical analysis”.

Author Response

Dear Respected Reviewer,

Thanks for providing us with your valuable comments. All the comments were addressed using point to point clarification in the attached file.

Thanking you

Regards

Round 2

Reviewer 3 Report (New Reviewer)

Dear editorial team,

Dear authors, 

Thank you very much for the revision of your initial version. The recommendations have been implemented excellently. From my point of view, nothing stands in the way of publishing the manuscript.

This manuscript is a resubmission of an earlier submission. The following is a list of the peer review reports and author responses from that submission.

Round 1

Reviewer 1 Report

The manuscript entitled: 

"Vitamin D, Cholesterol and DXA values as predictors for bi-maxillary bone mineral density"

Dear authors,

Title: Please correct the spelling:  bi-maxillary. Also, CONSIDER either Sentence case or capitalize each word.

Please add the email address to each author

Abstract:

Remove headings

rewrite the objective without the “null hypothesis”

Material and methods: specify inclusion and exclusion criteria rather than “enrollment of the patients those who satisfied the inclusion criteria”

remove “Pearson correlation test was tested to assess the correlation coefficient value among the measured parameters”

Conclusion:

please rewrite. It is unclear: “Vitamin D, cholesterol levels and Z values in women and men from young adulthood to middle age years to do appear to predict, over the long-term, subsequent bone mineral density in elderly years”

Keywords – write all in small caps

Introduction

please put the. after []

Please add the reference to cholesterol and its role in the craniomaxillofacial area

Define the aim better and do not use abbreviations before explaining them

The inclusion criteria should be better detailed

How did you select the 187 patients?

Were the researchers blinded?

At what level did you record bone mineral density on CBCT? Maxilla, mandible? incisors, molars?

What were the intra-examiner reliability was confirmed and Kappa values?

Please detail the statistical analysis

Results

You should describe better table 1,2,3,4,5,6 from a clinical perspective

It seems like tables were put there without explanation or connection between the aim and results

Discussion

Is too short and not related to the results

Please compare your own data with the ones from the literature

Add limitations and strengths of the manuscript

The conclusion should be rewritten. It is not supported by the research

References are in journal style.

Author Response

Dear sir,

Thanks for reviewing our manuscript. We appreciate the comments put forth by the reviewer and in response to that all comments are being addressed in the revised manuscript file.  Point to point clarification file is provided with the response to reviewer comments and changes made in the manuscript with its exact location.

Regards

Reviewer 2 Report

1. The authors mention sample size to be 187, but the number in not tallying with table 1 which is 177.

2. Authors do not mention anything about sample size calculation.

3.  Normal values of vitamin D ,cholestrol , BMD values. not mentioned in the study , changes wrt male and females, age group.

4. Discussion is vaguely written , not many points related to the study clearly explanied.

5. What age group is middle age individuals ?

Author Response

(The authors gave the same response as above.)

Round 2

Reviewer 1 Report

Dear authors,

Congratulations on your work!